# Novel Ran-RCC1 Inhibitory Peptide-Loaded Nanoparticles Have Anti-Cancer Efficacy In Vitro and In Vivo

**DOI:** 10.3390/cancers11020222

**Published:** 2019-02-14

**Authors:** Yusuf A. Haggag, Kyle B. Matchett, Robert A. Falconer, Mohammad Isreb, Jason Jones, Ahmed Faheem, Paul McCarron, Mohamed El-Tanani

**Affiliations:** 1Department of Pharmaceutical Technology, Faculty of Pharmacy, University of Tanta, Tanta 31111, Egypt; youssif.hagag@pharm.tanta.edu.eg; 2School of Pharmacy and Pharmaceutical Sciences, Saad Centre for Pharmacy and Diabetes, Ulster University, Cromore Road, Coleraine, Co. Londonderry BT52 1SA, UK; p.mccarron@ulster.ac.uk; 3Northern Ireland Centre for Stratified Medicine, School of Biomedical Sciences, C-TRIC, Altnagelvin Hospital Campus, Ulster University, Glenshane Road, Derry/Londonderry BT47 6SB, Northern Ireland, UK; k.matchett@ulster.ac.uk; 4Institute of Cancer Therapeutics, Faculty of Life Sciences, University of Bradford, Bradford BD7 1DP, UK; r.a.falconer1@bradford.ac.uk (R.A.F.); J.R.Jones@bradford.ac.uk (J.J.); 5School of Pharmacy and Clinical Sciences, University of Bradford, Bradford BD7 1DP, UK; M.Isreb1@bradford.ac.uk; 6Department of Pharmacy, Health and Well-being, University of Sunderland, Sunderland SR1 3SD, UK; ahmed.faheem@sunderland.ac.uk; 7Imhotep Diagnostics and Therapeutics, Europa Tool House, Springbank, Industrial Estate, Dunmurry BT17 0QL, Northern Ireland, UK; 8School of Chemistry and Biosciences, University of Bradford, Bradford BD7 1DP, UK

**Keywords:** Ran-RCC1 peptide, Ran, nanoparticle, breast cancer, lung cancer, anti-cancer, anti-metastatic, drug delivery

## Abstract

The delivery of anticancer agents to their subcellular sites of action is a significant challenge for effective cancer therapy. Peptides, which are integral to several oncogenic pathways, have significant potential to be utilised as cancer therapeutics due to their selectivity, high potency and lack of normal cell toxicity. Novel Ras protein-Regulator of chromosome condensation 1 (Ran-RCC1) inhibitory peptides designed to interact with Ran, a novel therapeutic target in breast cancer, were delivered by entrapment into polyethylene glycol-poly (lactic-co-glycolic acid) PEG-PLGA polymeric nanoparticles (NPs). A modified double emulsion solvent evaporation technique was used to optimise the physicochemical properties of these peptide-loaded biodegradable NPs. The anti-cancer activity of peptide-loaded NPs was studied in vitro using Ran-expressing metastatic breast (MDA-MB-231) and lung cancer (A549) cell lines, and in vivo using Solid Ehrlich Carcinoma-bearing mice. The anti-metastatic activity of peptide-loaded NPs was investigated using migration, invasion and colony formation assays in vitro. A PEG-PLGA-nanoparticle encapsulating *N*-terminal peptide showed a pronounced antitumor and anti-metastatic action in lung and breast cancer cells in vitro and caused a significant reduction of tumor volume and associated tumor growth inhibition of breast cancer model in vivo. These findings suggest that the novel inhibitory peptides encapsulated into PEGylated PLGA NPs are delivered effectively to interact and deactivate Ran. This novel Ran-targeting peptide construct shows significant potential for therapy of breast cancer and other cancers mediated by Ran overexpression.

## 1. Introduction

Metastasis is the major cause of cancer-related morbidity and mortality. Approximately 7 million people die from cancer-related cases per year and it is predicted that there will be more than 16 million new cancer patients every year by 2020 [1].

The eukaryotic cell Ran is a small GTPase of the Ras superfamily, which has been shown to be essential for directing nucleocytoplasmic transport, mitotic spindle fiber assembly and post-mitotic nuclear envelope dynamics [2]. Recently, several studies have shown that the overexpression of Ran in fibroblasts induces cellular transformation and tumor formation in mice [3]. In particular, Ran has been reported to be overexpressed in a wide range of human tumor types and cancer cell lines [4]. Moreover, overexpression of Ran was also shown to stimulate anchorage-independent growth in a non-invasive mammary cell line, cell attachment and invasion through Matrigel^®^ in vitro, and metastasis in syngeneic rats in vivo. Ran also functions as a novel effector of osteopontin-mediated malignant transformation. RAN GTPase (RAN) is one of the genes whose expression is substantially increased in relation to overexpression of osteopontin [5,6]. Moreover, Ran has been shown to be a potential therapeutic target for cancer treatment [7].

Silencing of Ran expression induces significantly greater apoptosis in cancer cells when compared to normal cells [8]. Cancer cells with mutations and/or abnormal expression in proto-oncogenes and/or suppressors with hyperactivation of the PI3K/Akt/mTORC1 and Ras/MEK/ERK pathways, which are the most frequently dysregulated signaling pathways in human cancers, have been shown to be more susceptible to Ran knockdown than normal cells [7].

The GTPase nature of Ran gives it the natural ability to behave as a switch in the cell, alternating its functions in response to the conversion between its GTP and GDP nucleotide-bound states [9,10]. These dichotomous states of Ran are divided spatially within the cell with higher RanGTP in the nucleus and more RanGDP in the cytoplasm. The different conformations of nucleotide-bound Ran are created and maintained due to the localization of different Ran regulatory proteins in the two different cell compartments. RanGTP is formed in the nucleus through interaction with the Guanine Nucleotide Exchange Factor RCC1 (Regulator of Chromosome Condensation 1) [11]. This interaction increases the rate of exchange of GDP for GTP on Ran and because the concentration of GTP is roughly 10 times greater than that of GDP in the cell, Ran is heavily loaded with GTP [12]. The catalytic activity of RCC1 is intensified through its bound position on the core nucleosome histones H2A and H2B [13], and interestingly the chromatin association of RCC1 is regulated by its interaction with Ran.

Ran is a soluble protein assembled from 216 amino acids with canonical sequence motifs and has a relatively low molecular size of 25 kDa. In order to block Ran-GTP formation and hence Ran activation, we have generated different inhibitory peptides designed to disrupt the Ran GDP-RCC1 interaction. We have identified three sites of interest for the generation of inhibitory peptides to the Ran GDP-RCC1 interaction, and then designed a family of inhibitory peptides including a contiguous sequence ranging from 6 to 25 amino acids in length (Patent Application GB1607593.9). Design of these peptides depends on the basic secondary structural analysis of the Ran protein, together with the protein crystallography of the complex between Ran and RCC1, and biochemical studies of the Ran-RCC1 interaction [14]. Three synthetic peptides (N-terminal (NT 3–12), C-terminal (CT 196–210) and Med (M 124–135) peptides) and three deleted peptides (ICT-24, ICT-25 and ICT-26 peptides) were used to interact with the pre-determined three sites of interest on the Ran structure. While these inhibitory peptides appear effective in silico, binding where predicted, their activity is reduced in vitro due to poor cellular permeability. The major challenge associated with effective delivery of anti-cancer therapeutic peptides is the difficulty in reaching the deep subcellular site of action. Peptide hydrophilicity restricts traversing of biological membranes. Furthermore, stability is poor due to rapid digestion by cellular proteolytic enzymes [14]. These challenges in peptide and protein drug delivery are not uncommon [15].

There are many examples of peptide therapeutics in the clinic, and many more currently in clinical development. Leuprolide, used in the treatment of prostate cancer, is an excellent example [16]. Peptides have already been used to manipulate critical regulatory networks in cancer cells, through targeting specific intracellular proteins required for tumour cell proliferation and invasion, or inhibition of essential signaling molecular function in cancer cells [17]. These peptides can be used for monotherapy or in combination with other conventional chemotherapeutic agents to maximise and extend the duration of therapeutic action [18]. Selected inhibitory peptides can extend the spectrum of other drugs already used to interfere with the function of a given gene product, including antisense oligonucleotides (siRNA and shRNA), intracellular antibodies, dominant-negative proteins and RNA aptamers [17].

As discussed, peptides present several challenges to their therapeutic application. Their undesirable physicochemical properties, such as variable solubility, low bioavailability and limited stability restrict their suitability for systemic delivery [19,20]. However, this is not an insurmountable challenge.

One approach to the delivery of peptides is via encapsulation and/or nanoparticle-based technologies. Polymeric nanoparticles (NPs), with their flexibility in formulation design, can improve the physicochemical, pharmaceutical and pharmacological properties of free peptides without altering their chemical structure and pharmacological activity [21]. Polymeric NPs enhance the therapeutic efficacy of peptides by achieving targeted delivery to cellular and intracellular sites of action through improving cellular uptake [22]. Polymeric NPs serve as a means of protecting therapeutic peptides from degradation and prolonging half-lives in vivo, while also providing the opportunity to control and sustain peptide release over a prolonged period of time, thereby fine-tuning and improving pharmacokinetic profiles [14,23,24]. Poly (lactic-co-glycolic acid) (PLGA) is an attractive polymeric substrate for use in encapsulation. It is biocompatible, biodegradable and is approved by various regulatory bodies for parenteral drug delivery applications [25]. In addition, PLGA has been shown to be capable of promoting the cellular uptake of peptides into the cytoplasm within 10 min of incubation [26]. 

The first aim of this study was to prepare a drug delivery system for novel Ran-GTPase inhibitory peptides, designed specifically to interact with and to deactivate RanGTP, through encapsulation into different polymeric NPs made from PLGA and its diblock copolymers of 5% and 10% PEG-PLGA. The second aim was to investigate the anti-cancer and anti-invasive action of the optimized NPs on breast cancer (MDA-MB-231) and lung cancer (A549) cells in vitro, both of which have been characterized with Ran overexpression. Anti-cancer efficacy in vivo was demonstrated using the Solid Ehrlich Carcinoma (SEC) xenograft model induced in mice.

## 2. Results and Discussion

### 2.1. Formulation and Characterization of Peptide-Loaded NPs

Peptide-loaded PLGA NPs were formulated using the water/oil/water emulsion solvent evaporation technique. Physicochemical parameters, such as mean particle size, zeta potential, encapsulation efficiency and in vitro release profile were determined. Pegylation of the PLGA backbone was used to adapt nanoparticulate properties and achieve a suitable size, high peptide loading efficiency and controlled therapeutic action.

The physicochemical characterisation of different peptide-loaded NPs (F1–F9) (Table 1) fabricated with three different polymers is illustrated in (Figure 1A–D). PEGylated NPs (F4–F9) prepared from 5 and 10% PEG-PLGA diblock copolymers, respectively, were significantly lower in size (*p* value < 0.05) than the nanoparticles prepared from PLGA polymer (F1–F3) (Figure 1A). PLGA NPs (F1–F3) exhibited higher negative zeta potential values compared to the PEGylated NPs (Figure 1B). PEGylation resulted in a significant increase (*p* value < 0.05) in peptide loading and entrapment efficiency (Figure 1C). This is explained by the unique proprieties of PEGylated polymers. PEG-PLGA as an amphiphilic polymer, which modifies the association of polymer molecules during nanoparticulate formation, eventually resulting in formation of smaller nanoparticles [24]. The presence of PEG chains on the nanoparticulate surface shields the peripheral charge that is prominent on PLGA-only NPs, which explains the reduction in zeta potential (Figure 1B) [27]. PEGylated polymers can form micelles more effectively than the PLGA because of its well-defined lipophilic portion remaining inside the micelles, and its hydrophilic component (PEG) projected to the outside. This creates a suitable environment for encapsulating hydrophilic molecules, like peptides, preventing untoward diffusion to the external aqueous phase and giving rise to high loading efficiencies [24]. 

The in vitro release profile of F1–F9 showed that peptide-loaded NPs prepared with PEGylated polymers displayed a large initial burst release followed by a gradual release profile, but the peptide-loaded PLGA NPs showed a smaller burst effect and then a lag phase characterised by a slow peptide release phase [24] (Figure 1D). The burst release mechanism is associated with surface-bound peptide molecules releasing rapidly, which then readily diffuse into the incubation medium. 

The data in Figure 1 clearly show that increasing the PEG content in the polymeric structure of PLGA NPs leads to a reduction in mean diameter to approximately 200 nm and an improvement in peptide loading. This observation was consistent across the three peptides used in this work. Subsequently, in further studies, the 10% PEG-PLGA particle matrix was selected as the optimal particulate matrix. Figure 2 shows the scanning electron microscope (SEM) image of 10% PEG-PLGA NPs, which exhibited a monodisperse size range with a smooth spherical surface free from pores.

### 2.2. In Vitro Antitumor Activity

The Ran-mediated anti-cancer activity of peptide-loaded NPs on the lung cancer cell line A549 is shown in (Figure 3A–C). Cell viability was evaluated by MTT assay after 24, 48, 72 and 96 h of treatment at 2, 4 and 8 µM (peptide equivalent) of loaded NPs, and equivalent concentration of the respective free peptides. Dose-response curves were used to evaluate the drug concentration that caused a 50% growth inhibition (IC_50_). Results confirmed that dimethylsulphoxide (DMSO) and blank NPs exhibited no cytotoxicity on lung cancer cells at the concentrations used. Free Med peptide and C-terminal peptide and their loaded nanoparticulate counterparts exhibited no cytotoxic action on A549 cell lines. Similar results were observed in the breast cancer cells (Appendix A). Significantly, N-terminal peptide-loaded NPs showed a concentration-dependent anti-cancer effect, while the respective free peptide showed no effect. N-terminal peptide-loaded NPs reduced cell viability and sustained cytotoxic action for up to four days after treatment. 

The average IC_50_ value for N-terminal peptide-loaded NPs for A549 cells was approximately 8 µM, which was achieved within 24 h. From these data, it is clear that N-terminal peptide-loaded NPs showed a significantly higher tumor growth inhibition (and lower resulting cell viability) in both cell lines. We concluded that the N-terminal peptide-loaded NPs sustained the cytotoxic action of the free drug due to enhanced cellular permeability. The lower cell viability and greater tumor growth inhibition compared to the free peptide was clearly observed after 96 h of treatment. Free peptides achieved no cytotoxic action due to low cellular permeability. Med peptide and C-terminal peptide-loaded NPs exhibited no cytotoxic action on both cell lines, even when delivered in nanoparticulate form. This might be attributed to a lack of interaction between the released peptides and RCC1 or Ran GDP. However, the N-terminal peptide-loaded NPs achieved significant anti-cancer activity represented by higher potential for tumor growth inhibition compared to other peptide-loaded NPs, suggesting effective interaction with RCC1 to prevent its role in Ran GDP activation to Ran GTP. These results are in good agreement with previous studies, [28] which confirmed that the RCC1 N-terminal tail controls its interaction with chromatin, which subsequently regulates its interaction with Ran. This interaction can be also modified by α-amino methylation of RCC1 N-terminal tail [29].

The antitumor activity of deleted peptide-loaded NPs encapsulating (ICT-24, ICT-25 and ICT-26 peptides) compared to full length N-terminal peptide-loaded NPs on the breast cancer cell line MDA-MB-231 is shown in Figure 4A–C. Cell viability was evaluated by MTT assay after 24, 48, 72 and 96 h of treatment at 2, 4 and 8 µM (peptide equivalent) of deleted and full length peptide-loaded NPs. ICT-26 and ICT-25 peptide-loaded NPs exhibited a minor cytotoxic effect on breast cancer cell lines compared to full length N-terminal peptide-loaded NPs. However, ICT-24 peptide-loaded NPs achieved significant cytotoxic action when compared to the full-length N-terminal peptide-loaded NPs. This is likely to be due to sharing the same active motif, which might be responsible for a possible interaction between the deleted peptide and RCC1 preventing its role in Ran GDP activation to Ran GTP. Treatment with scrambled peptide and scrambled peptide-loaded NPs showed no cytotoxicity to MDA-MB-231 cells (Appendix A). Furthermore, treatment of the MCF-10a cell line with full length N-terminal peptide and its loaded NPs showed no evidence of cytotoxicity, proving the safety of free peptide and peptide-loaded NPs (Appendix A). 

### 2.3. In Vitro Anti-Metastatic Activity

#### 2.3.1. Cell Migration Assay

An in vitro scratch assay was conducted on both MDA-MB-231 and A549 cell lines to investigate the effect of different doses of free N-terminal peptide and N-terminal peptide-loaded NPs on the migration capacity of two metastatic cell lines. Migration of adherent cancer cells is the translocation of these cells from one location to another. Migration of cancer cells from the primary tumor site to other sites is a predominant feature of metastatic MDA-MB-231 and A549 cancer cells [30]. Results in Figure 5A,B show the quantification of the degree of scratch closure in breast and lung cancer cell lines relative to the DMSO and blank nanoparticulate controls.

Treatment with free peptide showed no decrease in relative scratch closure for all concentrations in both cell lines. N-terminal peptide-loaded NPs showed a significant reduction compared to the free peptide treatments. A reduction of 42.2%, 66.2% and 76.5% in the scratch closure occurred after treatment with 2, 4 and 8 µM of N-terminal peptide-loaded NPs within 72 h in breast cancer cells. Similar to previous results, lung cancer cells treated with N-terminal peptide-loaded NPs showed a significant reduction of 46.8%, 60.6% and 66.8% in the degree of scratch closure compared to the free peptide after 72 h of using 2, 4 and 8 µM, respectively. These results demonstrated that N-terminal peptide-loaded NPs decrease the migratory capacity of invasive lung and breast cancer cells. N-terminal peptide-loaded NPs sustained and improved the anti-migration capacity of free peptide, which is essential to prevent metastasis of invasive breast and lung cancer cells. 

#### 2.3.2. Cell Invasion Assay

An in vitro invasion assay was conducted on lung and breast cancer cell lines to investigate the effect of different doses of free peptide and N-terminal peptide-loaded NPs on cell invasiveness. Dissemination of tumor cells from the primary site to distant locations starts with cell detachment followed by local invasion of the normal tissues adjacent to the tumor then infiltration through the lymphatic drainage system [30]. Treatment with N-terminal peptide-loaded NPs inhibited the invasion of both invasive MDA-MB-231 and A549 cancer types. Results in Figure 6A,B show the degree of invasion of treated breast and lung cancer cell lines through a Matrigel**^®^** membrane relative to the cells treated with DMSO and blank nanoparticulate controls. Treatment with N-terminal peptide-loaded NPs resulted in a significant decrease in cell invasion compared to free peptide for all concentrations in both cell lines. In MDA-MB-231 cells, after 72 h of treatment, reductions of 35.3%, 45.7% and 62.8% in invasiveness were observed after treatment with 2, 4 and 8 µM of N-terminal peptide-loaded NPs, respectively. Furthermore, N-terminal peptide-loaded NPs showed a significant reduction compared to free peptide treatments in A549 cells. Reductions of 36.6%, 48.5% and 59.1% in cell invasion after 72 h of treatment were observed with 2, 4 and 8 µM of N-terminal peptide-loaded NPs, respectively. These results showed that N-terminal peptide-loaded NPs inhibit invasiveness of lung and breast cancer cells. N-terminal peptide-loaded NPs sustained and improved the anti-invasive efficacy of the free peptide, which is essential for inhibiting metastasis of invasive breast and lung cancer cells.

#### 2.3.3. Colony Formation Assay

A colony formation assay was used to study the long-term anti-cancer activity of free peptide and N-terminal peptide-loaded NPs [31]. The number of colonies formed after 7 days was estimated for cells treated with DMSO and blank NPs (controls), and cells treated with different concentrations of free peptide and N-terminal peptide-loaded NPs, as shown in Figure 7A,B. Treatment with N-terminal peptide-loaded NPs resulted in significantly lower numbers of colonies when compared to free peptide. Breast cancer cells treated with N-terminal peptide-loaded NPs demonstrated 34.6%, 44.4% and 58.6% reductions in number of colonies formed after treatment with 2, 4 and 8 µM of N-terminal peptide-loaded NPs, respectively, for 7 days. Moreover, treatment with N-terminal peptide-loaded NPs resulted in a significant reduction in colony formation in A549 cells. 

Lung cancer cells treated with N-terminal peptide-loaded NPs showed only 28.4%, 35.2% and 62.8% reductions in colony formation after one week of treatment with 2, 4 and 8 µM of N-terminal peptide-loaded NPs, respectively. These results showed that treatment with N-terminal peptide-loaded NPs provided a sustained and long-term inhibitory effect on cancer cell growth and its proliferation when compared to treatment with free peptide.

#### 2.3.4. Ran Activation Assay

Ran regulates the spatial and temporal coordination of mitotic spindle formation by activation of Ran-GDP to Ran-GTP by interaction with its guanine exchange factor, regulator of chromosome condensation (RCC1). We have previously shown that peptides delivered from loaded NPs (peptide-NP) significantly inhibit the interaction between Ran-GDP and RCC1 [14]. This result showed that Ran peptide-NPs inhibited the formation of the active form of Ran, Ran-GTP (Figure 8).

### 2.4. In Vivo Anti-Tumor Activity 

Recently, the role of Ran-GTPase in acquisition of certain cancer features, such as limitless replicative potential, evasion of apoptosis and tissue invasion, has been clarified [6]. This manifests itself mainly through errors in mitotic functions that can leave cells vulnerable to genetic instability, which as previously mentioned, is a risk factor, promoting the development of the hallmarks of cancer [2].

The Solid Ehrlich Carcinoma (SEC) xenograft model induced in mice was established as a valid model that is commonly used to investigate different chemotherapeutic treatment strategies for breast cancer [32]. This model reflects a high degree of malignancy due to its high virulence ability, rapid development and highly infiltrative nature [33]. Previous studies showed that Ehrlich ascites carcinoma (EAC) cells were characterized by Ran-GTP over expression [34,35,36]. Therefore, it can be used as a potential model to study the effect of anti-Ran GTPase blockade peptides in vivo.

Compared to the control Group I and Group II (treated with 10 mg kg^−1^ free peptide), there was a significant decrease in the tumor volume of Group III mice (treated with 10 mg kg^−1^ of N-terminal peptide-loaded NP) at all recording points after 18 days from starting the treatment. Macroscopic pictures showed a significant reduction in tumor volume following treatment with peptide-loaded NPs when compared to treatment with free peptide (Figure 9A,B). Tumor volumes of Group II exhibited an insignificant decrease at all recording points of the experiment (Figure 9C). The % tumour growth inhibition (TGI) in Group III was 20% after 18 days of treatment and gradually increased to 42% at the end of the experiment (Figure 10).

Histopathological examination of the studied groups revealed the typical picture of Solid Ehrlich carcinoma (Figure 11A–E). Examination of sections prepared from the tumor tissue of the control Group I showed anaplastic malignant ductal undifferentiated carcinoma cells with pleomorphic changes, increased nucleo/cytoplasmic ratio and perineural invasion (Figure 11A, H&E ×400). Sections prepared from Group II (treated with 10 mg kg^−1^ of free peptide) showed anaplastic malignant ductal carcinoma cells with pleomorphism, increased nucleo/cytoplasmic ratio with skeletal muscle invasion and necrosis. The cells were spherical in shape, containing relatively large, highly chromatophilic nuclei with one or more prominent nucleoli. Giant tumor cells are also seen (Figure 11B, H&E ×400). 

Histopathological examination of Group III (treated by 10 mg kg^−1^ of N-terminal peptide-loaded NPs) revealed significantly different profiles. Two specimens showed necrotic malignant cells in the center surrounded by a rim of viable tumor cells and a collar of inflammatory cellular filtrate and fibroblastic proliferation (Figure 11C, H&E ×100). Three sections showed mononuclear cellular infiltrate as well as macrophage infiltration of the necrotic tumor tissue in partial responding samples (Figure 11D) (H&E ×400). The other five samples showed complete absence of either viable or necrotic tumor tissue with macrophages and mononuclear cellular infiltrate and fibrosis (Figure 11E). 

The Ran GDP-GTP cycle regulates multiple cellular processes that contribute to initiation and progression of breast cancer, which is primarily apoptosis, invasion and angiogenesis. The mechanisms by which Ran overexpression could impact on the course and survival in cases of breast cancer have been reported [8]. Disruption of the Ran cycle by blocking the interaction between Ran GDP and RCC1 by subcellular delivery of an anti-Ran GTPase inhibitory peptide resulted in Ran deactivation by preventing Ran GTP formation. N-terminal peptide-loaded NPs successfully delivered the peptide which interfered with the RCC1-Ran-GDP interaction that perturbs the GTP/GDP cycle causing defects in mitotic spindle morphology, including misalignment of chromosomes and abnormal numbers of spindle poles [28].

A major challenge associated with anti-cancer therapeutics, and peptide therapeutics in particular, is a failure to reach desired cellular and intracellular target sites, while minimizing action at nonspecific sites. Surface modification of NPs with PEG has emerged as a strategy to prolong circulation time, minimize non-specific uptake and allow for specific tumor-targeting through the enhanced permeability and retention effect (EPR). The rapid uptake of PLGA nanoparticles by the macrophages of the reticulo-endothelial system (RES), primarily in the liver and spleen, could be reduced significantly by modifying their surface with PEG. The presence of PEG chains on the surface can protect NPs from capture by macrophages and improves cytoplasmic transport and reduces possible enzymatic degradation [37]. 

Treatment by N-terminal Ran GTPase inhibitory peptide-loaded NPs resulted in a significant reduction in tumor volume and a marked increase in the % TGI. Histopathological studies showed partial and complete response to nanoparticulate-based treatment compared to a nil response following treatment with free peptide. These results confirmed the successful delivery of peptide-loaded NPs to the tumor site, taking advantage of the EPR effect, where the Anti-Ran GTPase inhibitory peptide can be released into the subcellular site to block Ran GTP formation and therefore inhibit its role in tumorigenesis.

## 3. Materials and Methods 

### 3.1. Materials

Ran-GTP peptides (N-terminal peptide, C-terminal peptide and Med peptide) was synthesized by GL-Biochem Ltd. (Shanghai, China). Deleted peptides (ICT-24, ICT-25 and ICT-26) were synthesized at the Institute of Cancer Therapeutics, University of Bradford, Bradford, UK, using conventional solid phase peptide synthesis. The sequences of three deleted peptides were CAQGEPQVQFK (ICT-24), EPQVQFK (ICT-25) and VQFK (ICT-26). PLGA, PLGA-co-PEG block copolymers, Poly (vinyl alcohol), Resomer® RGP d 50105 and phosphate buffered saline (PBS) were obtained from Sigma (St. Louis, MO, USA). A Lowry Protein Assay kit was obtained from Thernofisher Scientific (Waltham, MA, USA).

### 3.2. Preparation of Peptide-Loaded NPs

A modified, double emulsion, solvent evaporation method [24,38] was employed in this study. Anti-Ran-GTPase inhibitory peptides were dissolved in an aqueous solvent (0.2 mL of PBS) to form the internal water phase and mixed with dichloromethane (DCM, 2 mL) containing 5% w/v of different polymers, then emulsified by using a Silverson L5T Homogenizer (Silverson Machines, Chesham, UK) at 6000 rpm for 2 min. The primary emulsion was prepared as we described previously [14]. The final product was stored in a desiccator at ambient temperature. Different nanoparticulate formulations, together with their identifying codes, are listed in Table 1.

### 3.3. Physico-Chemcial Characterization of Peptide-Loaded NPs

#### 3.3.1. Particle Size and Zeta Potential

Particle size and distribution of peptide-loaded NPs were determined using a dynamic light scattering principle technique (Zetasizer 5000, Malvern Instruments, Malvern, UK). A suspension of NPs (5 mg mL^−1^) was vortexed for 5 min. An aliquot from this suspension was diluted in ultrapure water before measurement of mean diameter (in triplicate). Dynamic light scattering and electrophoretic mobility under an applied electric field was used to measure zeta potential. Peptide-loaded NPs were suspended in 0.001 M KCl with an appropriate adjustment in conductivity. The average of 3 measurements (Zetasizer 5000, Malvern Instruments) was reported [39].

#### 3.3.2. Particle Surface Morphology

Surface morphology was characterized by scanning electron microscopy (FEI Quanta 400 FEG, FEI Company, Hillsboro, OR, USA). A sample of NPs was mounted on carbon tape and sputter-coated with gold under vacuum in an argon atmosphere prior to imaging.

#### 3.3.3. Determination of Peptide Loading and Encapsulation Efficiency

Peptide content was determined by both direct extraction from lyophilized NPs and by an indirect procedure based on determination of non-encapsulated peptide [14]. The encapsulation efficiency of the peptides was evaluated as we described previously [14]. The indirect method determined non-encapsulated peptide content in the supernatant using reversed phase chromatography [14]. Peptide encapsulation was calculated from the difference between the initial amount of peptide added and the non-entrapped peptide remaining in the supernatant after nanoparticulate fabrication. Each sample was assayed in triplicate and the average values using the two different assay method results were combined as the percentage peptide encapsulation efficiency.

#### 3.3.4. In Vitro Release Studies

A sample of 0.5 mg of the peptide-NP was diluted in PBS to a concentration of 0.5 mg/mL. The resultant suspension was incubated in a reciprocal shaking water bath at 37 °C and 100 rpm. Samples were withdrawn at time intervals 1, 12, 24, 48, 72, 96, 120, 144, and 168 h and the volume was readjusted using fresh release media. Samples were centrifuged at 22,000× *g* and the concentration of the peptide was measured using High Performance Liquid Chromatography (HPLC). The stability of the peptide during the release experiment was previously verified [14].

### 3.4. Cell Culture

MDA-MB-231 cells (metastatic breast cancer) and A549 cells (metastatic lung cancer) were generously provided by Prof. El-Tanani (Institute of Cancer Therapeutics, University of Bradford, Bradford, UK). These cells were maintained as monolayer cultures in Dulbecco’s Modified Eagle’s Medium–High Glucose (DMEM-Hi) medium (Gibco BRL, Grand Island, NY, USA) supplemented with 10% fetal bovine serum (Gibco BRL, Grand Island, NY) and 1% penicillin–streptomycin (Gibco BRL) at 37 °C in a humidified atmosphere (5% CO_2_).

### 3.5. In Vitro Antitumor Assay

Cytotoxicity assays were performed as previously reported [40] with minor modifications. MDA-MB-231 and A549 cells (5 × 10^4^ cells/well in 500 µL media) were seeded in 24-well plates. The following day, A549 cells were treated with different concentrations of three different Ran GTPase blockade peptides and their NPs suspended in Optimem^®^ media. MDA-MB-231 cells were treated by N-terminal, C-terminal and Med peptide-loaded NPs and deleted peptides-loaded NPs (ICT-24, ICT-25 and ICT-26) suspended in the same media. Medium containing equivalent amounts of DMSO or blank NPs was used as the control. The media were changed every 2 days and no further dose of free peptide or peptide-loaded NPs was added. The cytotoxic effects of 2, 4 and 8 µM of the free peptide or peptide-loaded NPs were determined every 24 h using an MTT assay to assess cell viability. Treated cells were washed with 500 µL PBS, then 500 µL of 15% MTT dye solution in complete media was added to each well. The plates were incubated at 37 °C and 5% CO_2_ for an additional 3 h. The supernatant was discarded and formazan crystals solubilized by adding 500 µL of DMSO and the solution vigorously mixed to dissolve the precipitate. The color intensity was measured at 570 nm (reference wavelength 630 nm) in a microplate reader (Fluostar Omega, BMG Lab Tech GMbH, Bucks, UK). The anti-proliferative effects of different doses of free peptide or peptide-loaded NPs treatments were calculated as a percentage of cell growth with respect to the DMSO and blank NPs controls. The absorbance of the untreated cells was set at 100%. All the experiments were repeated three times.

### 3.6. Migration Assay

The effect of peptide constructs on tumor cell migration was performed was assessed using an in vitro scratch assay [41]. Cells (7.5 × 10^5^) were suspended in 2 mL media and seeded into each well of a 6-well plate. The cells were pipetted drop by drop around the edge of each well and shaken gently to distribute the cells evenly in each well. The plates were placed in incubator to allow cells to grow to confluence. A scratch was introduced to confluent cells using a sterile pipette tip by carefully scraping down the center of each well. Old media were removed and 2 ml of each drug concentration containing 2, 4 and 8 µM of the free N-terminal peptide or N-terminal peptide-loaded NPs suspended in Optimem^®^ media was added. Medium containing equivalent amounts of DMSO or blank NPs were used as controls. A photograph for each wound at 0 h was taken and recorded. A photograph was taken at the same time of day at 24 h, 48 h, and 72 h. To quantify the results from photographic data, ImageJ software was used to measure scratch width at 3 different locations and the average was recorded. The degree of closure over 24 h was calculated by the difference between the width at 0 h and 24 h for each treatment concentration. The degree of closure for each treatment was normalized to the degree of closure in the DMSO control. All the experiments were repeated three times.

### 3.7. Invasion Assay

An invasion assay was performed as previously described [42]. This investigated the effect of peptide-loaded nanoparticulate treatment on the ability of MDA-MB-231 and A549 cells to invade across a Matrigel^®^-coated membrane. 8 µm membrane transwell inserts were placed into the wells of 12-well plates (BD Bioscience, San Jose, CA, USA). The membrane transwell was coated with Matrigel^®^ before seeding the cancer cells. Matrigel^®^ was placed in ice and left to thaw at 4 °C, then diluted 1:1 with water. 135 µL of the solution was placed carefully into the required number of inserts. The inserts were shaken gently to ensure the Matrigel^®^ solution spread evenly on the membrane. The whole plates were left in the laminar flow hood to dry for 60 min. Cells (4 × 10^4^) were seeded into each insert in a suspension of 200 µL of serum-free media. Next, 100 µL containing 2, 4 and 8 µM of the free N-terminal peptide or N-terminal peptide-loaded NPs suspended in serum-free media was added to each insert. The same volumes of 100 µL containing equivalent amounts of DMSO or blank PLGA NPs were used as controls. Finally, 700 µL of DMEM media containing 10% FBS was added to each well to bathe the outer membrane surface and act as chemoattractant. The next day, to new 12-well plates, 500 µL of methanol was added to the corresponding number of wells used. The media in both chambers was removed and cotton wool scrubbers were used to clean off any remaining cells or Matrigel^®^ layer. The inserts were placed into methanol for 10 min for fixation. Once fixation was complete, the inserts were left to dry. Using a scalpel, the membranes were cut out and placed with the original outer surface facing upwards in wells of a new 12-well plate. Crystal violet (250 µL) was added to each well to stain cells that invaded through the membrane pores. The plates were shaken for 20 min. Afterwards, the crystal violet was pipetted out and the membranes washed thoroughly in their wells with Milli-Q^®^ water. The membranes were left to air dry for 60 min. Ethanol solution (250 µL of 70%) was added to membranes in each well and shaken for 30 min. Finally, 200 µL was pipetted out from each well and placed into the wells of 96-well plate. Absorbance at 590 nm was measured using a microplate reader (Fluostar Omega, BMG Lab Tech GMbH). The effects of different doses of free peptide and peptide-loaded PLGA NPs treatments were calculated as a percentage of cell invasions with respect to the DMSO and blank nanoparticluate controls. The absorbance of the untreated cells was set at 100%. All the experiments were repeated three times.

### 3.8. Colony Formation Assay

For the colony formation assay, 500 cells of MDA-MB-231 and A549 cells were seeded in 2 mL media in 6-well plates and allowed 2 days to attach and initiate colony formation [31]. These cells were treated with different concentrations of 2, 4 and 8 µM of the free N-terminal peptide or N-terminal peptide-loaded NPs suspended in Optimem^®^ media over a period of 7 days. The plates were washed twice with PBS, fixed in chilled methanol, stained with crystal violet, washed with water and air dried. The number of colonies was counted by using a magnifying lens. The percent of colony formation was calculated by dividing the number of colonies formed in treatment by the number of colonies formed in DMSO or blank NPs.

### 3.9. Ran Activation Assay

Lung cancer cell line A549 was used for this experiment. 1.5 × 10^5^ cells/2 mL complete growth medium were seeded into 6-well tissue culture plate 24 h prior to the experiment. A Ran activation assay kit (Cell Biolabs, San Diego, CA, USA) was used to measure the extent of Ran activity, as described previously in [14].

### 3.10. In Vivo Antitumor Activity

The antitumor activity of peptide-loaded NPs was evaluated in vivo in mice-bearing solid tumors. An Ehrlich Ascites Carcinoma (EAC) cell line was obtained from the Experimental Oncology Unit of the National Cancer Institute (NCI), Cairo University, Egypt. Cancer cell viability was 98% as judged by the trypan blue exclusion assay. The xenograft model of Solid Ehrlich Carcinoma (SEC) was induced in female Swiss albino mice using 2 × 10^6^ viable EAC cells suspended in 0.2 mL isotonic saline. EAC cells were aspirated from the peritoneal cavity of mice, washed with saline and implanted subcutaneously between thighs of the lower limb of each mouse. The tumor was developed in 100% of the mice, with palpable solid tumor masses (≥100 mm^3^) achieved within 12 days post-implantation [43,44]. 

#### 3.10.1. Animal Groups and Treatment Protocol 

Animal procedures in this study complied with UK Animal Use in Research Guidelines (UK Directive) and European Union Guidelines (European Directive 2010/63/EU) on animal Experimentation for the Protection and Humane use of Laboratory Animals. Experiments were conducted at an accredited Animal Experimentation Facility (MBC, Queen’s University, UK). Procedures were approved by the Ethics Committee of Queen’s University, Belfast. Thirty adult female Swiss albino mice with a weight range of 18–20 g were allowed *ad libitum* water and standard pellet chow (EL-Nasr Chemical Company, Cairo, Egypt) through the whole period of the in vivo experiment. Mice were housed and allowed to become acclimatized to laboratory conditions for 7 days prior to the experiment. The in vivo experiment was conducted in accordance with the National Institute of Health guide for the care and use of Laboratory animals (NIH Publication No. 8023, revised 1978). All mice were rendered tumor-bearing and divided randomly into 3 equal groups; Group I served as control group and received a vehicle of isotonic saline. The tumor-bearing mice in Group II and Group III were treated 2 days per week by intraperitoneal injection of free peptide and peptide-loaded NPs in a dose of 10 mg kg^−1^, respectively. The treatment by either saline or peptide-loaded NPs was started from Day 12 to Day 38 post-implantation. 

#### 3.10.2. Tumor Volume (V) and Percentage Tumor Growth Inhibition (% TGI) 

Tumor volumes were recorded from the start point at Day 12 post-implantation and thereafter every 2 days till the last record at Day 38 post-implantation, just prior to scarification of survivors. Using a Vernier caliper, the following formula was used to calculate the volume of the developed tumor mass [45]: Tumor volume (mm^3^) = 0.52 × length × (width)^2^. Drug efficacy was expressed as the percentage tumor growth inhibition calculated as % TGI = 100 − (T/C × 100), where T is the mean relative tumor volume (RTV) of the treated groups and C is the mean RTV in the control group. RTV is defined as V_x_/V_1_, where V_x_ is the tumor volume at each point of the experiment before mice scarification and V_1_ is the tumor volume at the start point of the treatment [46].

#### 3.10.3. Processing of Tumor Tissue Samples 

At the end point of the experiment (Day 38 post-implantation), all survivors were sacrificed. The tumor was excised, washed immediately with ice-cold saline and the specimen preserved in 10% formalin solution. After xylene treatment, the specimens were embedded in paraffin blocks. Five-micron thick sections were cut and stained with hematoxylin and eosin (H&E). H&E stained sections were examined and investigated. 

### 3.11. Statistical Analysis

Statistical analysis was conducted using GraphPad Prism 5.0 software (GraphPad, San Diego, CA, USA). In vitro results of particle size, zeta potential, entrapment efficiency and values of in-vitro release profiles were presented as mean ± (SD). Results of in vitro cell based assays and in vivo results were presented as mean ± (SEM). All results were treated statistically using one-way analysis of variance (ANOVA) followed by post-hoc Tukey test. *p* value < 0.05 was considered statistically significant.

## 4. Conclusions

The involvement of Ran in the development and progression of breast cancer has been described to be due to its complex mechanisms enabling tumorigenesis and metastasis. An optimized nanoparticle formulation has shown significant cytotoxic effects in metastatic cancer (MDA-MB-231 and A549) cells compared to free peptide. Cellular drug uptake and permeability of peptide-loaded nanoparticles *versus* release profile of N-terminal peptide will be investigated in a follow-up study. *N*-terminal peptide-loaded NPs have also shown enhanced anti-metastatic effects in cancer cells to inhibit the migration, invasion and colony formation capacity of metastatic cancer cells. 

The anti-cancer and anti-metastatic potential of the free Ran blockade peptides was significantly improved through a nanoparticulate delivery system, which significantly enhanced anti-cancer activity. In vivo results confirmed the anti-cancer activity of the peptide-loaded NPs, achieving successful peptide delivery to the tumor and 42% tumor growth inhibition. The Ran-RCC1 blockade peptide blocks the dynamic interaction of RCC1 with Ran, thereby preventing the transformation of inactive RanGDP to active RanGTP. Since this shuttling between the two different forms of nucleotide bound Ran is essential for its activity, the prevention of this process should modulate Ran function. In conclusion, Ran-inhibitory peptide-loaded PEG-PLGA NPs were shown to be promising anti-cancer and anti-metastatic agents that could have applications in therapeutic oncology.

## Figures and Tables

**Figure 1 cancers-11-00222-f001:**
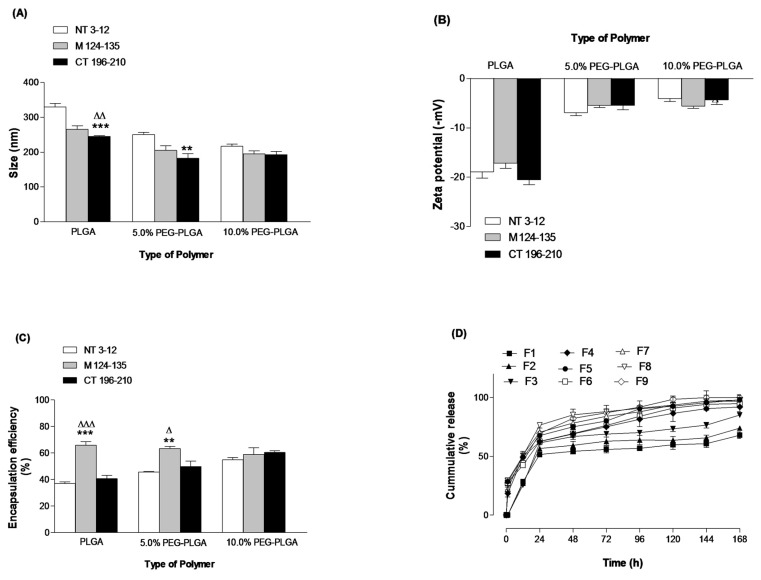
Effects of polymer type on size (**A**), zeta potential (**B**), encapsulation efficiency (**C**) and in vitro release (**D**). Values are mean ± SD with *n* = 3. For 1A–C, ** *p* < 0.01, *** *p* < 0.001 compared with NT 3–12. ^Δ^
*p* < 0.05, ^ΔΔ^
*p* < 0.01, ^ΔΔΔ^
*p* < 0.001 compared with M 124–135.

**Figure 2 cancers-11-00222-f002:**
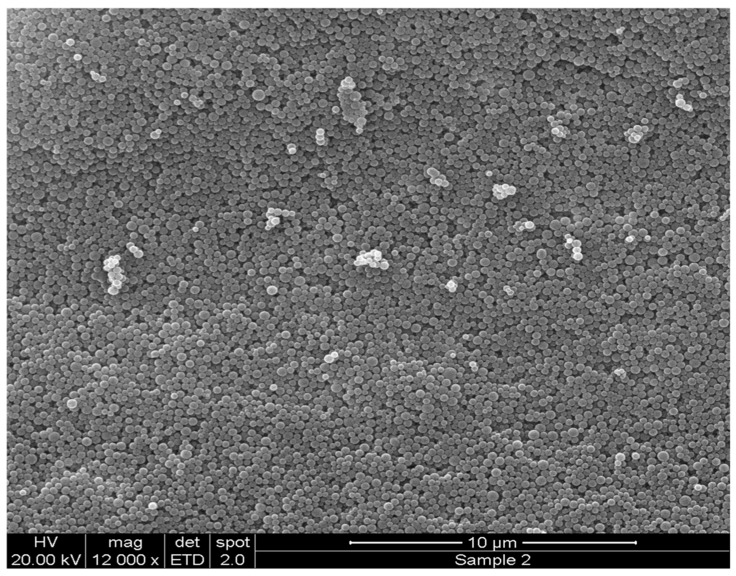
Scanning Electron Microscope (SEM) images of NT 3–12 peptide-loaded 10% PEG-PLGA NPs (F7) after formulation.

**Figure 3 cancers-11-00222-f003:**
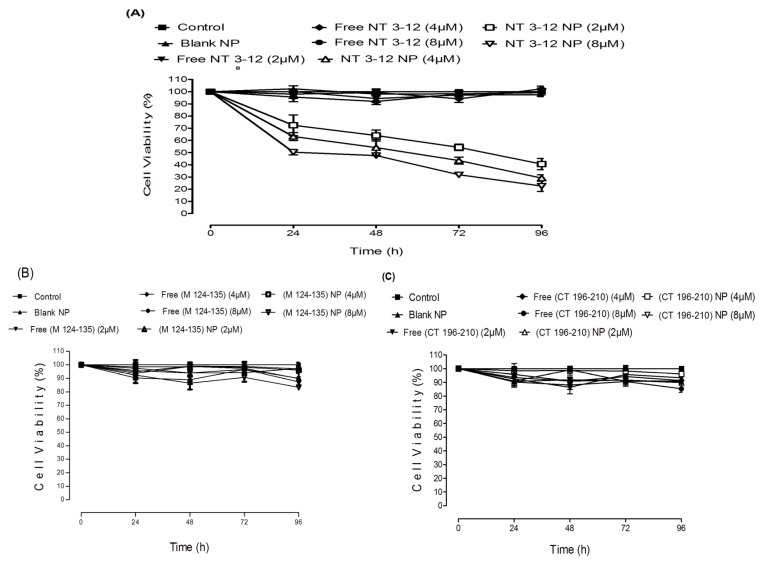
A549 cell viability results of different doses of NT 3–12 peptide and NT 3–12 peptide-loaded NPs (**A**), M 124–135 peptide and M 124–135 peptide-loaded NPs (**B**) and CT 196–210 peptide and CT 196–210 peptide-loaded NPs (**C**) after 24, 48, 72 and 96 h.

**Figure 4 cancers-11-00222-f004:**
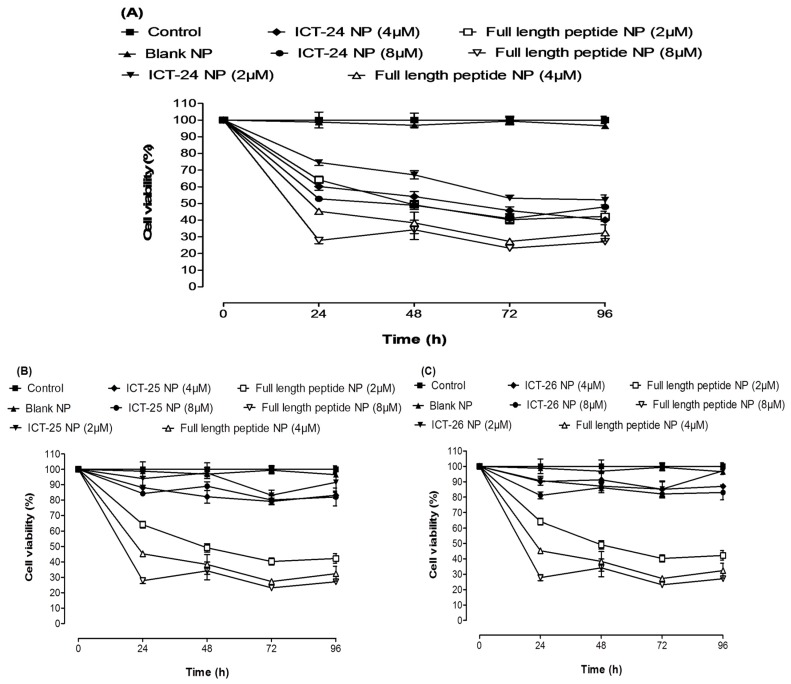
MDA-MB-231 Cell viability results of different doses of deleted peptide NPs and full length NT 3–12 peptide-loaded NP. ICT-24 peptide-loaded NPs (**A**), ICT-25 peptide-loaded NPs (**B**) and ICT-26 peptide-loaded NPs (**C**) after 24, 48, 72 and 96 h.

**Figure 5 cancers-11-00222-f005:**
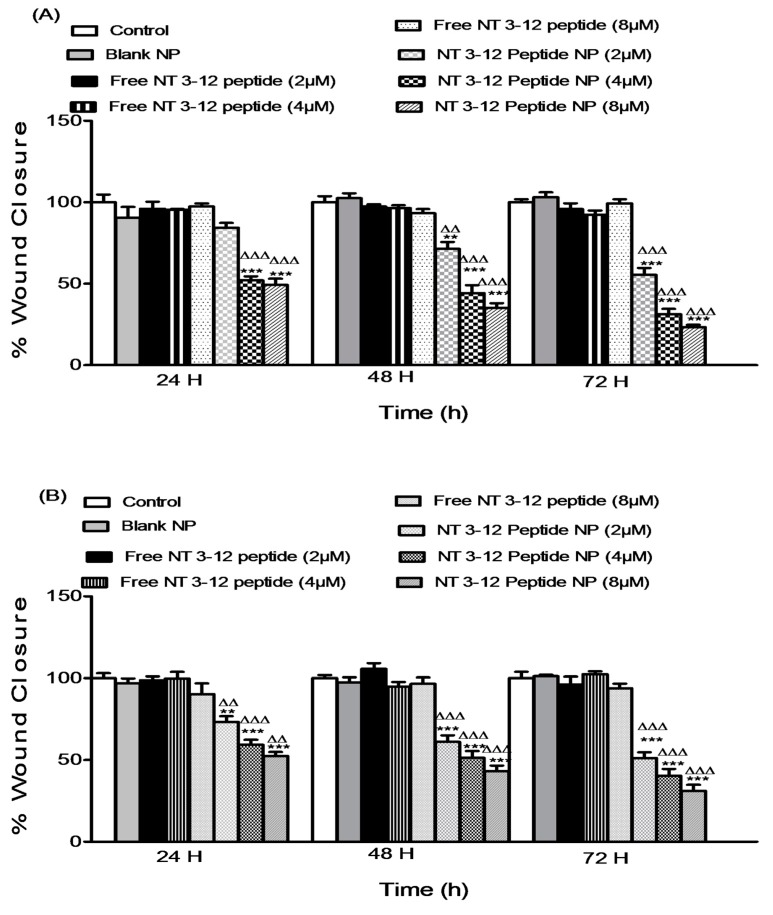
Effect of different doses of free N 3–12 peptide and N 3–12 peptide-loaded NPs on MDA-MB-231 migration (**A**) and A549 migration (**B**). Values are mean ± SEM with *n* = 3. ** *p* < 0.01, *** *p* < 0.001 compared with Control and blank NP. ^ΔΔ^
*p* < 0.01, ^ΔΔΔ^
*p* < 0.001 compared with the same dose of free peptide.

**Figure 6 cancers-11-00222-f006:**
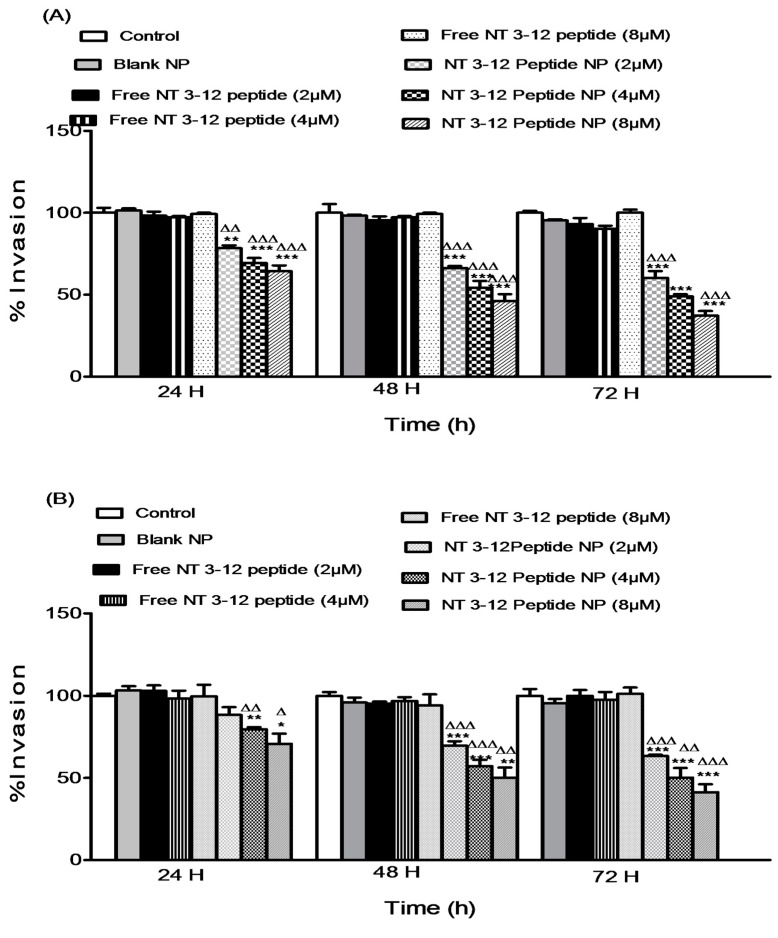
Effect of different doses of free N 3–12 peptide and N 3–12 peptide-loaded NPs on MDA-MB-231 invasion (**A**) and A549 invasion (**B**). Values are mean ± SEM with *n* = 3. * *p* < 0.05, ** *p* < 0.01, *** *p* < 0.001 compared with Control and blank NP. ^Δ^
*p* < 0.05, ^ΔΔ^
*p* < 0.01, ^ΔΔΔ^
*p* < 0.001 compared with the same dose of free peptide.

**Figure 7 cancers-11-00222-f007:**
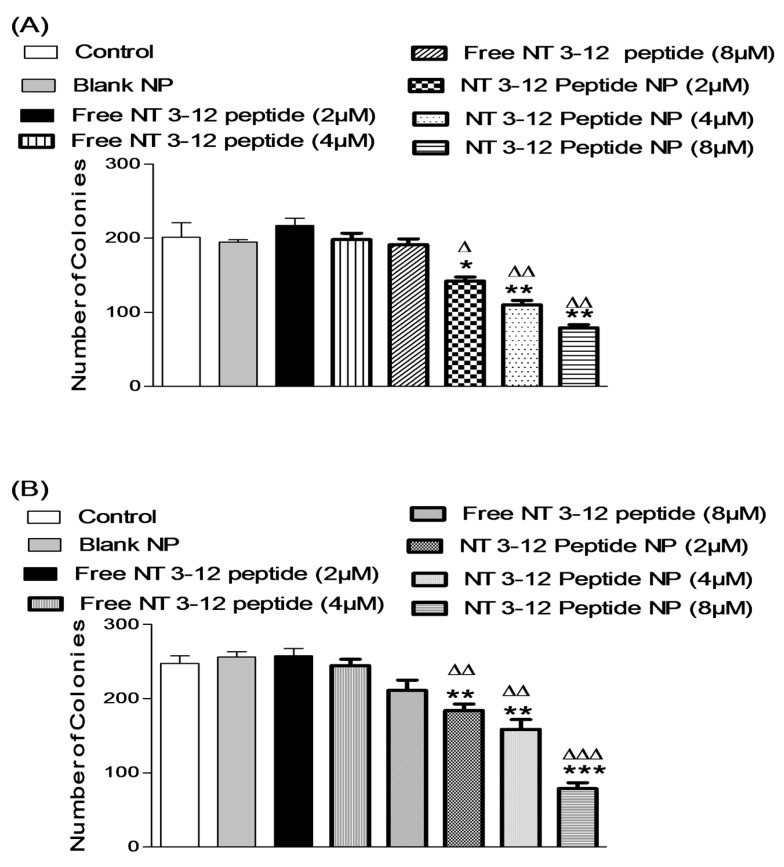
Effect of different doses of free N 3–12 peptide and N 3–12 peptide-loaded NPs on MDA-MB-231 number of colonies (**A**) and A549 number of colonies (**B**). Values are mean ± SEM with *n* = 3. * *p* < 0.05, ** *p* < 0.01, *** *p* < 0.001 compared with Control and blank NP. ^Δ^
*p* < 0.05, ^ΔΔ^
*p* < 0.01, ^ΔΔΔ^
*p* < 0.001 compared with the same dose of free peptide.

**Figure 8 cancers-11-00222-f008:**
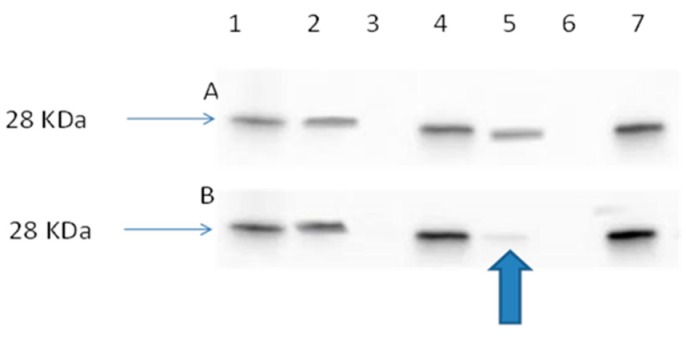
(**A**) Immunoblot image of lung cancer cell line (A549), control cells and (A549) cell line treated with blank 10% PEG-PLGA NP. Immunoblot positive control, (lane 1). A549 control cell lysate, (Lane 2). A549 control cell lysate after spiking with GDP (worked as negative control), (Lane 3). A549 control cell lysate after spiking with GTPγS (worked as positive control), (Lane 4). A549 cell lysate after treatment with blank NP, (Lane 5). A549 cell lysate after treatment with blank NPs and spiking with GDP, (Lane 6). A549 cell lysate after treatment with blank NPs and spiking with GTPγS, (Lane 7). (**B**) Immunoblot image of A549 cells treated with free NT 3–12 peptide and A549 cells treated with NT 3–12 peptide-loaded NPs. Immunoblot positive control, (lane 1). A549 cell lysate after treatment with NT 3–12 free peptide, (Lane 2). A549 control cell lysate after treatment with NT 3–12 free peptide and spiking with GDP (worked as negative control), (Lane 3). A549 control cell lysate after treatment with NT 3–12 free peptide and spiking with GTPγS (worked as positive control), (Lane 4). A549 cell lysate after treatment with NT 3–12 peptide-loaded NPs, (Lane 5). A549 cell lysate after treatment with NT 3–12 peptide-loaded NPs and spiking with GDP, (Lane 6). A549 cell lysate after treatment with NT 3–12 peptide-loaded NPs and spiking with GTPγS, (Lane 7).

**Figure 9 cancers-11-00222-f009:**
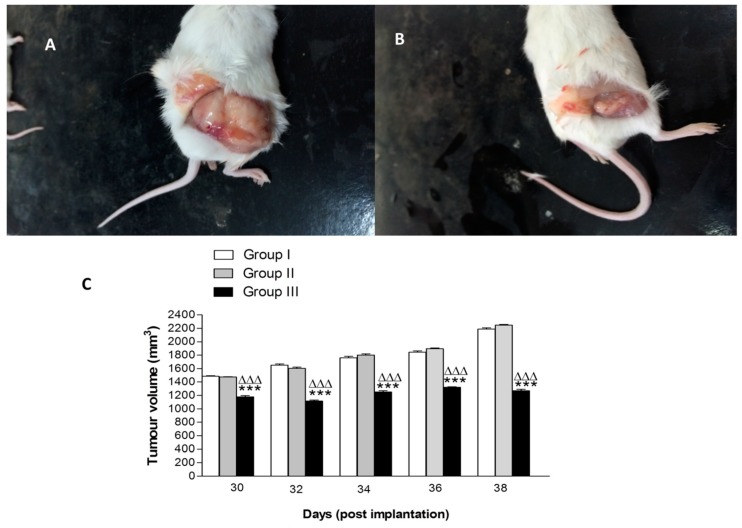
Effect of treatment with free N 3–12 peptide and N 3–12 peptide-loaded NPs on tumour volume in SEC mice. Photograph of Group II treated with free peptide (**A**) and photograph of Group III treated with N 3–12 peptide-loaded NPs (**B**). Tumor volume of groups at the recording points every 2 days from Day 12 (start point of treatment) to the last record at Day 38 post-implantation (**C**). For 9C, values are mean ± SEM with *n* = 10. *** *p* < 0.001 compared with Group I (control group). ^ΔΔΔ^
*p* < 0.001 compared with Group II (treated by free peptide 10 mg/kg).

**Figure 10 cancers-11-00222-f010:**
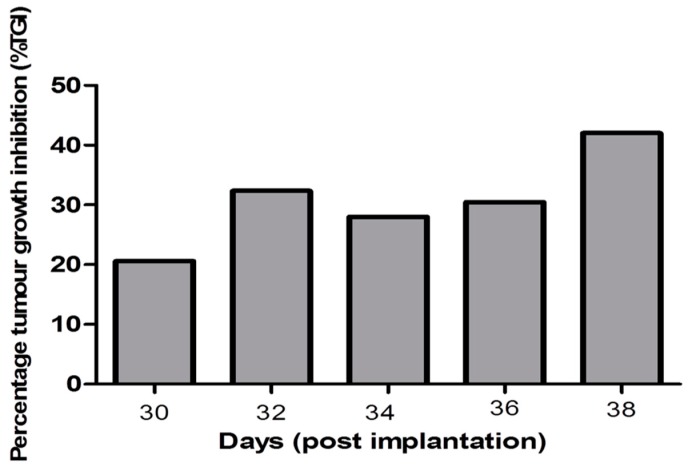
Percentage tumor growth inhibition (% TGI) in Group III (treated by NT 3–12 peptide-loaded NPs 10 mg kg^−1^) relative to Group I (control group).

**Figure 11 cancers-11-00222-f011:**
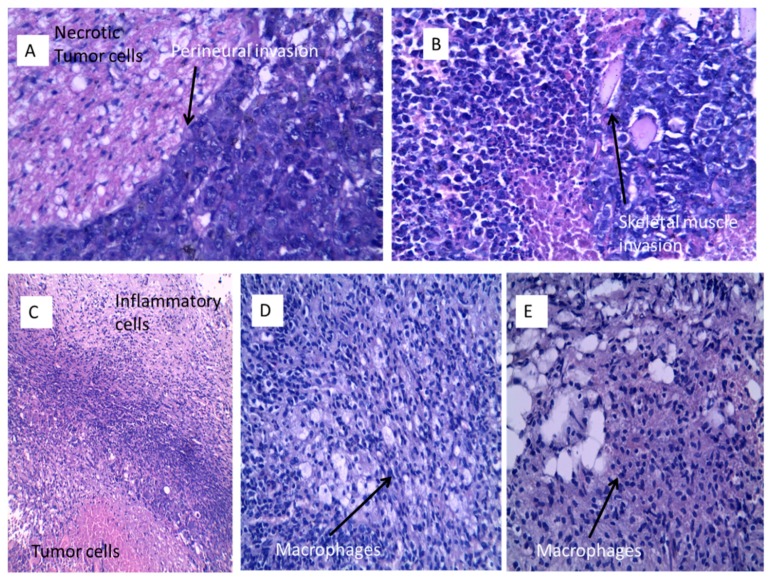
Histopathological findings of SEC sections stained with H&E. Group I (**A**) showing large necrotic centers surrounded by undifferentiated carcinomatous cells (H&E ×400). Group II (**B**) showing the cellular details of the tumor, the cells are spherical in shape, containing relatively large, highly chromatophilic nuclei with one or more prominent nucleoli, giant tumor cells are also seen (H&E ×400). Group III (**C**) showing viable and necrotic tumor cells surrounded by a layer of inflammatory cells (H&E ×100), (**D**) showing partial responding samples with macrophages and mononuclear cellular infiltrate (H&E ×400) and (**E**) showing complete absence of viable or necrotic tumor cells with macrophages and lymphocytes infiltration (H&E ×400).

**Table 1 cancers-11-00222-t001:** Different formulations of peptide-loaded NPs and corresponding identifiers.

Formula ID	Polymer Type	Encapsulated Peptide *
F1	PLGA	NT 3–12
F2	PLGA	M 124–135
F3	PLGA	CT 196–210
F4	5% PEG-PLGA	NT 3–12
F5	5% PEG-PLGA	M 124–135
F6	5% PEG-PLGA	CT 196–210
F7	10% PEG-PLGA	NT 3–12
F8	10% PEG-PLGA	M 124–135
F9	10% PEG-PLGA	CT 196–210

* NT 3–12 is coded for N-terminal peptide, CT 196–210 is coded for C-terminal peptide and M 124–135 is coded for Med peptide.

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
