# Peer review of "Novel Ran-RCC1 Inhibitory Peptide-Loaded Nanoparticles Have Anti-Cancer Efficacy In Vitro and In Vivo"

_cancers, 2019, doi:10.3390/cancers11020222_

Round 1

Reviewer 1 Report

Tumor delivery of novel Ran-RCC1 inhibitory peptide using nanoparticles is quite interesting considering the challenges of systemic delivery of naked peptides. There are, however, some major issues that should be addressed to prove the clarity of the experimental findings:

Why was intraperitoneal injection other that intravenous administration used?

What is the release mechanism of the encapsulated peptide from the particles? Does the release happen before or after internalization in target tumor cells?

Only 3 mice were used per group, which was not  an ideal experimental strategy. Based on our experience, the standard deviation (SD) in tumor regression study is substantially high. But, in this work, the SD was shown to be too small.

Biodistribution (pharmacokinetic) study is required to know the fate of the delivered peptides.

Author Response

Dear Reviewers,

Thank you very much for your valuable comments and suggestions. All revised sections were highlighted by yellow colour in the uploaded manuscript. Please note that current Ran activity assay of figure 8 has been done using lung cancer cells A549. However, the previous published Ran activity in Ref 14 has been performed using breast cancer cells, MDA MB 231.

Thank you for your attention.

Best regards,

Prof Mohamed El-Tanani

Why was intraperitoneal injection other that intravenous administration used?

With the nanoparticles average size of 200nm, it was deemed after discussion that it is safer not to use the IV route to avoid possibility of mouse tail vien blockages.

What is the release mechanism of the encapsulated peptide from the particles? Does the release happen before or after internalization in target tumor cells?

We would like to thank the reviewer and we will persue this question in next  future study. We are planning the next study will  focusing on the pharmacokinetics and pharmacodynamics of the drug delivery system. In this study we focused on the effect of the drug in vitro and in vivo. However from the data presented in this study we would assume that a significant proportion of the drug release occur at the targeted tissue and within the cells.

Only 3 mice were used per group, which was not  an ideal experimental strategy. Based on our experience, the standard deviation (SD) in tumor regression study is substantially high. But, in this work, the SD was shown to be too small.

We have used 10 mice per group (section 3.10.1 of the methodology)

Biodistribution (pharmacokinetic) study is required to know the fate of the delivered peptides.

We agree with the reviewer 1 that this will be a useful study to peruse in future but unfortunately it was not part of the scope for this paper. It will be the focus of a future piece of work focusing on the kinetics and dynamics of the release from the drug delivery system

Reviewer 2 Report

In the paper titled "Novel Ran-RCC1 inhibitory peptide-loaded nanoparticles have anti-cancer efficacy in vitro and in vivo" Yusuf Haggag and co-authors investigated new Ran-RCC1 inhibitory peptode-loaded nanopaticles against tumor cells and in animal models. Authors reported antimetastatic peptide-loaded NPs and antitumor activity also confirmed in in-vivo breast cancer animal models. This research is novel and deserves to be publshed, but after a thoroughly revision taking in cosideration of following suggestions\corrections\improvements:

1) A moderate English correction\revision by mother tongue is suggested in order to remove few typos and imperfections

2) I would reccomend\suggest to add\provide further characterisation analysis data (eg AFM\SEM images of NPs) to further support this research

3) What was choice of cancer cells? Did author compare also with healthy cells?

4) NPs uptake\colocalisation mechanism and quantification (eg by FACS\Confocal?) seems to be ovelooked. Please discuss\provide data about this important point 

5) In vivo data: what was driving choice of animals (eg tiype of mices\female\male ect)? please discuss this issue.

6) An update of literature references taking into account different type of drugs, similar NPS\cells\animals is also suggested.

Author Response

Many thanks for taking the time to review the manuscript. Please find the answers below:

1) A moderate English correction\revision by mother tongue is suggested in order to remove few typos and imperfections

The mother tongue authors have revised the entire manuscripts before the resubmission.

The changes we did to the manuscript hopefully addressed this quesiton

2) I would reccomend\suggest to add\provide further characterisation analysis data (eg AFM\SEM images of NPs) to further support this research

SEM image is provided (Figure 2)

3) What was choice of cancer cells? Did author compare also with healthy cells?

The cancer cells used were breast cancer cells MDA-MB-231 and lung cancer A549 cells. We have used healthy normal cells MCF-10a as a control which showed the encapsulated peptide (NPs) had no toxitiy effect on cell (supplementary data Figure S3).

4) NPs uptake\colocalisation mechanism and quantification (eg by FACS\Confocal?) seems to be ovelooked. Please discuss\provide data about this important point

This is a very important topic indeed and it will be part of our future work that will focusing on the pharmacokinetics and pharmacodynamics of the drug delivery system.

5) In vivo data: what was driving choice of animals (eg tiype of mices\female\male ect)? please discuss this issue.

The majoradvantage of this model is that it may be more relevant to the development of human disease because the tumors reside in the tissue appropriate for the histotype. Ehrlich carcinoma is one of the commonest transplantable tumors that appeared firstly as a spontaneous breast cancer in a female mouse (1). Ehrlich carcinoma is an undifferentiated carcinoma that has high transplantable capability, no regression, rapid proliferation, shorter life span, 100% malignancy and also does not have tumor-specific transplantation antigen. Ehrlich carcinoma has a resemblance with human tumors which are the most sensitive to chemotherapy due to the fact that it is undifferentiated and that it has a rapid growth rate (2).

6) An update of literature references taking into account different type of drugs, similar NPS\cells\animals is also suggested. 

Please find the modification as requested in the revised manuscript.

Round 2

Reviewer 1 Report

There has been virtually no attempt to address my comments. New experimental data is required.

Author Response

We have addressed the questions raised by both referees. Referee 1 requested additional experiments, which we feel are beyond the scope of the current study. The academic editor reviewed these responses, and was happy that we had addressed the concerns. They asked for a comment to be inserted into the conclusion referring to future experiments, which we have done.

We therefore feel that we have addressed both referees comments, and those of the editor.

Reviewer 2 Report

Authors have satisfactorily addressed all issues raised by previous review and therefore manuscript is significatively improved to be acceptable for publication in Cancers in current revised version

Author Response

Thank you very much.